# Reconstructing the sediment concentration of a giant submarine gravity flow

Christopher John Stevenson [1], Peter Feldens[2], Aggeliki Georgiopoulou [3], Mischa Schönke[2], Sebastian Krastel [4], David J.W. Piper [5], Katja Lindhorst[4] & David Mosher [6]

Submarine gravity flows are responsible for the largest sediment accumulations on the planet, but are notoriously difficult to measure in action. Giant flows transport 100s of $km^3$ of sediment with run-out distances over 2000 km. Sediment concentration is a first order control on flow dynamics and deposit character. It has never been measured directly nor convincingly estimated in large submarine flows. Here we reconstruct the sediment concentration of a historic giant submarine flow, the 1929 "Grand Banks" event, using two independent approaches, each validated by estimates of flow speed from cable breaks. The calculated average bulk sediment concentration of the flow was 2.7–5.4% by volume. This is orders of magnitude higher than directly-measured smaller-volume flows in river deltas and submarine canyons. The new concentration estimate provides a test case for scaled experiments and numerical simulations, and a major step towards a quantitative understanding of these prodigious flows.

[1] School of Earth, Ocean and Ecological Sciences, University of Liverpool, Liverpool L69 3GP, UK. [2] Leibniz Institute for Baltic Sea Research, Warnemünde 18119 Rostock, Germany. [3] School of Earth Sciences, Science Centre-West, Belfield, Dublin, Ireland. [4] Christian-Albrechts-Universität zu Kiel, Institute of Geosciences, Otto-Hahn-Platz 1, 24118 Kiel, Germany. [5] Bedford Institute of Oceanography, Dartmouth, NS B2Y 4A2, Canada. [6] Center for Coastal & Ocean Mapping, 24 Colovos Road, Durham, NH 03824, USA. Correspondence and requests for materials should be addressed to C.J.S. (email: christopher.stevenson@liverpool.ac.uk)

Submarine gravity flows are mixtures of sediment and water. They are driven downslope by their excess density, which is generated from sediment suspended within the flow. They entrain ambient seawater at their head and upper interface, and either erode or deposit on the seabed. As the principal agent for transporting sediment across the Earth's surface, they have a significant influence on global sediment cycling and nutrient fluxes into the ocean[1], and are responsible for the largest sediment accumulations on the planet[2]. They also pose a major geohazard to seafloor infrastructure, such as telecommunication cables that carry >95% of global internet traffic[3], and oil and gas pipelines upon which our economies depend[4].

The concentration of sediment within a flow dictates almost all aspects of flow dynamics and style of deposition[5]. However, sediment concentration is a poorly constrained flow property. This is because there are few in situ measurements of natural submarine flows in action. These are restricted to small-volume (<$10^7$ m$^3$ of sediment) flows within slope canyons and fjord river deltas[6–13]. Therefore, our understanding of submarine flow dynamics is primarily inferred through analysis of their deposits, using insights from scaled physical experiments and numerical simulations.

Submarine gravity flows are vertically stratified, whereby a higher-concentration, coarser-grained lower layer is overlain by a thicker dilute upper layer transporting finer-grained material[6, 13–15]. In turn, the lower layer is continuously stratified with increasing sediment concentrations and grain sizes towards the bed[16]. When flows pass through confining topography, the character of their sediment deposits and the height to which they drape the confining slopes is a proxy for the vertical distribution of sediment in the flow[17–23], with the upper limit of erosion or deposition termed a trimline.

Previous work has used such constraints from medium scale ancient flows ($10^7$–$10^{10}$ m$^3$ of sediment) to reconstruct flow properties, yielding a wide range of potential depth-averaged sediment concentrations between 0.0007 and 2.5% vol.[18, 19, 21, 22, 24]. Crucially, these studies lack directly measured flow properties so that equations relating sediment concentration to dynamic flow properties, such as velocity, cannot be solved. In this case, flow velocities are approximated via the grain size of deposits[18, 21] or channel morphology[21, 24], which introduce considerable uncertainty. As a consequence, estimates of sediment concentration from flow deposits are wide-ranging and lack validity.

Large-volume submarine flows (>$10^{10}$ m$^3$ of sediment) are even less well-understood. These giant flows are highly destructive and transport vast amounts of sediment (some >100 km$^3$) over tens of thousands of square kilometres[25–27]. Such volumes of sediment eclipse the annual global river discharge of sediment into the ocean by an order of magnitude[25, 27]. The concentration of giant submarine flows has never been convincingly estimated due to virtually no directly measured flow properties and a lack of high-quality field data. This leaves our understanding fundamentally limited; rooted in qualitative interpretations.

Here we examine a classic historic giant submarine flow where velocity is known from timing of seafloor cable breaks[28]. Using new sediment cores and multibeam bathymetry, together with legacy submersible and core data[29], we establish the erosional trimline through a submarine channel network and then use these data to reconstruct the bulk sediment concentration of the flow.

## Results

**Field data**. Our study is based on the analysis of sediment cores, a submersible dive transect, and bathymetric and backscatter data

taken from deep-water offshore the Grand Banks, Newfoundland (Fig. 1a) (Supplementary Fig. 1). In 1929, a $M_W$ 7.2 earthquake triggered a large-sediment-volume (175 km$^3$) submarine flow and a tsunami that killed 28 people[30]. The flow travelled down slope through several channel systems, sequentially breaking seafloor cables in its path. The cable breaks provide a direct measure of frontal flow velocity at several locations along the flow pathway[28, 29, 31] (Fig. 1a).

Deposits of the 1929 event are recorded in the tops of sediment cores located in 4000–5000 m water depth across the Eastern Valley channel network (Fig. 2). The underlying stratigraphy is correlated between cores (Supplementary Figs 1 and 2)[32], which enables us to recognize the extent of erosion and deposition by the 1929 flow. Within the channel thalwegs and lower channel margins, 1929 deposits consist of an erosional surface >2 m deep overlain by thick (>1 m) structureless gravels or by a thin (5 cm) coarse-sand lag and sandy mud (Supplementary Fig. 2). Such deposits indicate a high-energy erosive flow with thin lags representing the majority of sediment that bypassed and was deposited farther downslope[30, 33]. Across inter-channel highs and upper parts of channel margins, 1929 flow deposits comprise dark brown, thin (up to 20 cm), fine sandy muds, which drape a regional olive grey hemipelagic mud (Supplementary Fig. 2). Hence, at these localities the 1929 flow is interpreted to have been non-erosive, low-energy and transporting only fine-grained sediments.

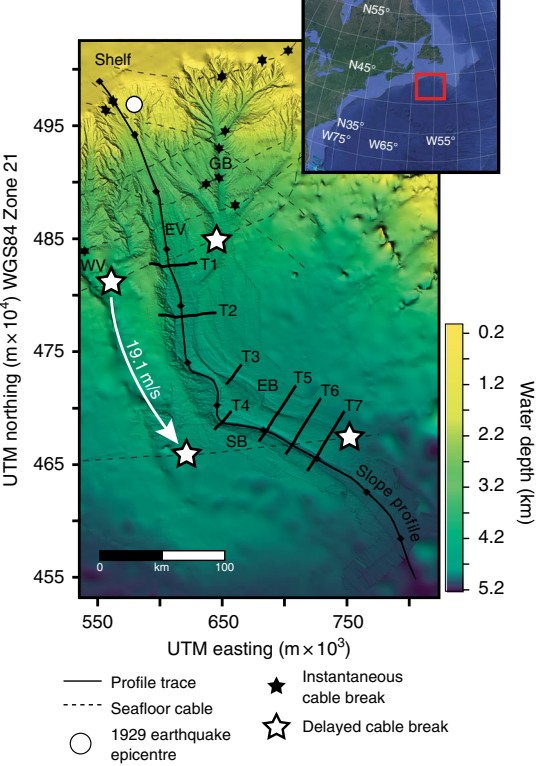

**Fig. 1** Bathymetry of the Grand Banks slope shown via a General Bathymetric Chart of the Oceans (GEBCO) base map, which is overlain by higher-resolution swath bathymetry collected aboard Cruise MSM47 in 2015. Several major channel systems were pathways for the 1929 flow: Western Valley (WV), Grand Banks Valley (GB), and Eastern Valley (EV), which splits into two smaller channels: East Branch (EB) and South Branch (SB). Delayed cable breaks provide a direct measure of flow speed (19.1 ms$^{-1}$). A down slope profile (Fig. 4) runs through the Eastern Valley channel system with 7 channel cross-section profiles (T1-T7) (Fig. 3). Insert map uses satellite imagery from Google Earth Pro$^{TM}$

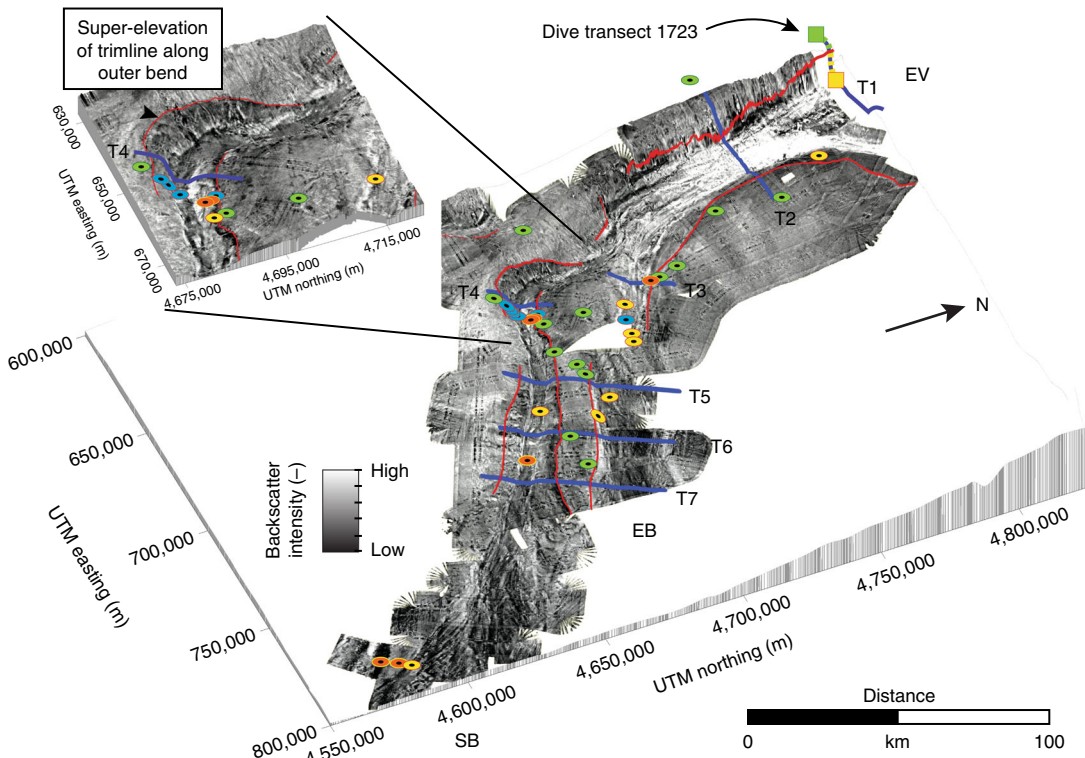

**Fig. 2** Acoustic backscatter from Cruise MSM47 (see 'Methods') across the lower parts of the Eastern Valley channel system with interpretation of erosional trimlines from the 1929 flow. The Eastern Valley (EV) splits into two smaller channels: East Branch (EB) and South Branch (SB). High-intensity backscatter represents rough sandy deposits whilst low-intensity represents smooth mud. Erosional trimlines are inferred from sharp boundaries in backscatter intensity running along the margins of the channels (red lines). Cores show major erosion by the 1929 flow within the channels that extends up to the elevation of the trimlines (gravels—yellow, bypass drapes—blue and bounced—orange circles). Undisturbed sediments occur above the trimline (green circles). Dive transect 1723 (squares) shows fresh outcrop and bio-erosion (yellow line) sharply changing into undisturbed sediments (green line). Channel transects shown as blue lines (T1–T7). The backscatter insert shows a more detailed picture of the super-elevated trimlines around the sharp bend in South Branch (T4)

Submersible dive 1723 started on the floor of the Eastern Valley and traversed up its western margin (Fig. 2). The boulder-strewn channel floor was cut by a 40 m deep scour at the foot of the valley wall[29]. The lower valley wall comprised fresh (angular) outcrop of mudstones. At 230 m above the valley floor the angularity ('freshness') of the outcrop decreased with a coincident appearance of immobile epifauna. This is interpreted to represent the erosional trimline from the 1929 flow[13].

Acoustic backscatter data show sharp boundaries between zones of high-backscatter and low-backscatter intensity that run along the channel margins (Fig. 2). Submersible dive observations and sediment cores ground-truth these backscatter boundaries in several places. The combination of well-compacted muds below the erosion surface and the sand deposits above it result in high acoustic backscatter, in contrast to the low backscatter from recently deposited hemipelagic and fine sandy muds. Thus the backscatter boundaries are interpreted as erosional trimlines of the 1929 flow. Their elevation above the channel thalwegs can be measured across the Eastern Valley channel network (T1–T7; Figs. 2 and 3).

**Reconstructing flow concentration**. We interpret the erosional trimlines to represent the thickness of the higher-concentration lower layer of the flow. The lower layer transported all the coarse-grained sediment load, which was responsible for most of the sediment concentration in the flow[5, 8, 17, 34]. The sandy mud deposits from shallower than the erosional trimlines are inferred to have been deposited from the overlying dilute, fine-grained,

upper-layer of the flow. These field data are now used to reconstruct the bulk concentration of the 1929 event.

We employ two independent approaches: downslope gravitational driving force and super-elevation of a flow around a bend. Assuming a uniform steady flow through a straight channel (T1–T3 and T5–T7; Fig. 2), flow thickness can be related to parameters of velocity, slope and sediment concentration through[18, 19, 21, 35]:

$$U^2 = \frac{RgCH_f \sin\theta}{C_d + E_w} \qquad (1)$$

where $R$ is specific density of sediment in seawater (1.53); $C$ is sediment concentration; $H_f$, the height of the velocity maximum from the bed (¼ of flow thickness)[36]; $\theta$, the downstream slope angle; $E_w$, the water entrainment coefficient across the upper flow interface ($0.072\sin\theta/1000$; see 'Discussion')[18], and $C_d$, the basal friction coefficient (estimated between 0.003 and 0.0045; see 'Methods')[37].

At a channel bend (T4), the following cross-flow momentum equation can be derived[18, 21, 37]:

$$\frac{p_t U^2}{Q} = g\Delta p \frac{dH}{dr} \pm p_t fU \qquad (2)$$

The three terms in Eq. 2 represent, from left to right, the centrifugal force, the pressure gradient and the Coriolis force, where $p_t$ is the density of the lower layer of the flow; $\Delta p$ is the flow excess density with respect to seawater; $U$, the depth-averaged downstream flow velocity (3.8  m s$^{-1}$; see 'Methods'); $g$,

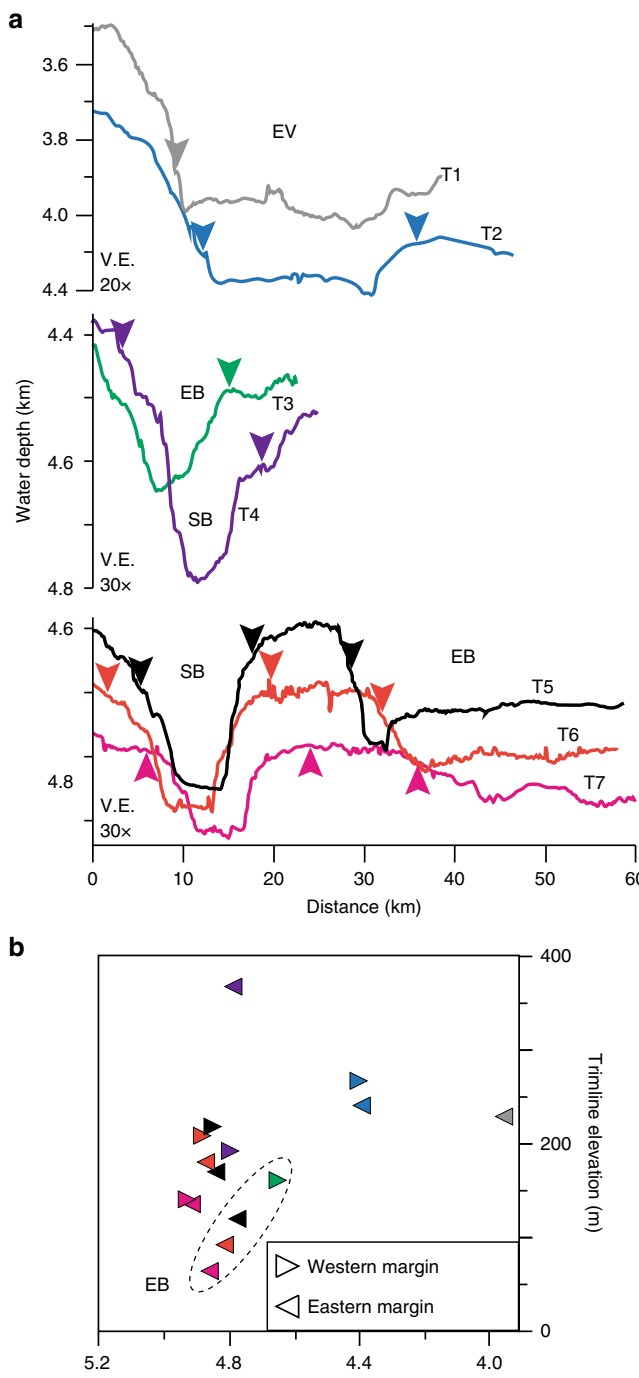

increases along the flow pathway (T1–T7) because the measured flow speed is an average value over a changing slope. This means that as the slope decreases, there must be a concomitant increase in sediment concentration to balance Eq. (1). Therefore, we estimate the bulk sediment concentration of the 1929 flow as an average of these values: 2.7% through South Branch and 5.4% through East Branch. As the deposit volume of the 1929 flow is constrained ($\sim$175 km$^3$)[30], a concentration of 2.7–5.4% indicates the duration of the flow through the Eastern Valley channel network was 4–8 h (see 'Methods').

## Discussion

There are assumptions and limitations to our approach that deserve consideration. Equation (1) is particularly sensitive to the input parameters governing basal friction ($C_d$) and upper surface water entrainment ($E_w$). These terms are not well-constrained by field data and hence introduce uncertainty to estimates of sediment concentration. The basal friction coefficient ($C_d$) is estimated using bed roughness (skin friction) derived from the grain sizes measured along the thalweg of the Eastern Valley (see 'Methods'). However, the 1929 flow has eroded and remoulded the Eastern Valley, meaning that it is difficult to assess exactly what sediments were present pre-1929. Upper and lower estimates of $C_d$ are used to reflect this uncertainty. The small range in values we use for $C_d$ (0.003–0.0045) results in error in flow concentration estimates of between 0.3–1.5% vol. The degree of error increases downslope due to decreasing flow thickness (Fig. 4).

Our formulation of water entrainment across the upper surface of the flow ($E_w$) is extrapolated from experiments[38, 39]. However, numerical simulations suggest entrainment in natural flows is likely to be 3–4 orders of magnitude less than predicted from experiments[40, 41], or so low that it would be considered negligible[34, 42]. As an approximation we reduce our upper surface entrainment value by 3 orders of magnitude, which typically decreases the estimates of sediment concentration by approximately 0.03% vol. (up to $\sim$1% vol.).

Our approach also assumes conservation of mass and neglects deposition or entrainment of sediment (erosion) along the flow's pathway. Evidence of erosion by the 1929 flow extends along the length of the Eastern Valley channel system (Fig. 2). Therefore, it is likely that a significant volume of sediment was added to the flow, which in turn, would progressively increase its sediment concentration downslope[43–46]. This trend is seen in our results and is probably, in part, a product of flow bulking from erosion (Fig. 4). However, the measured flow speed is an average over a changing slope, which manufactures a similar trend via Eq. (1). To resolve the relative contributions of these factors, more closely spaced measures of flow velocity would be required.

We estimate the lower-layer ($\sim$70–230 m thick; Fig. 3) of the 1929 flow as having a bulk sediment concentration between 2.7 and 5.4% vol. As gravity flows are vertically stratified, this bulk value represents a concentration gradient with lower concentrations in the upper parts of the flow and increasing sediment concentrations towards its base. Direct measurements of fine-grained (silt and fine-sand), dilute (<0.04% vol.) and low sediment volume submarine flows estimate near-bed sediment concentrations 3–12 times higher than depth-averaged values[6, 8, 9, 13]. Applying these gradients to our depth-averaged estimates from the 1929 flow suggests that near-bed sediment concentrations were likely to have been significantly higher than 10% vol. At these high sediment concentrations the flow becomes stratified into a concentrated near-bed grain flow layer that is sharply overlain by the overriding, more dilute, part of the flow[14, 47–50] (Fig. 5).

**Fig. 3** Cross-section topographic profiles and trimline elevations. **a** Channel cross-section topographic profiles T1–T9 with erosion trimlines marked (arrows) from the backscatter bathymetry (Fig. 2). **b** Plot showing trimline elevations from channel floor bathymetry (colour coded to T1–T9 profiles). EV Eastern Valley, EB East Branch, SB South Branch (see Fig. 1)

the acceleration due to gravity; $Q$, the channel-bend radius; dH/dr, the slope of the upper interface of the flow along the radial direction (dH, the height difference between inner-bend and outer bend trimline, dr, the width of the channel); and $f$, Coriolis acceleration ($f = 2\Omega\sin\Phi$, $\Phi = 42°$N, $\Omega$ is the angular speed of the Earth's spin).

Equations (1) and (2) yield bulk sediment concentrations for the flow between 0.5–4.8% through South Branch, and 0.5–10.2% through East Branch (Fig. 4). Flow concentration progressively

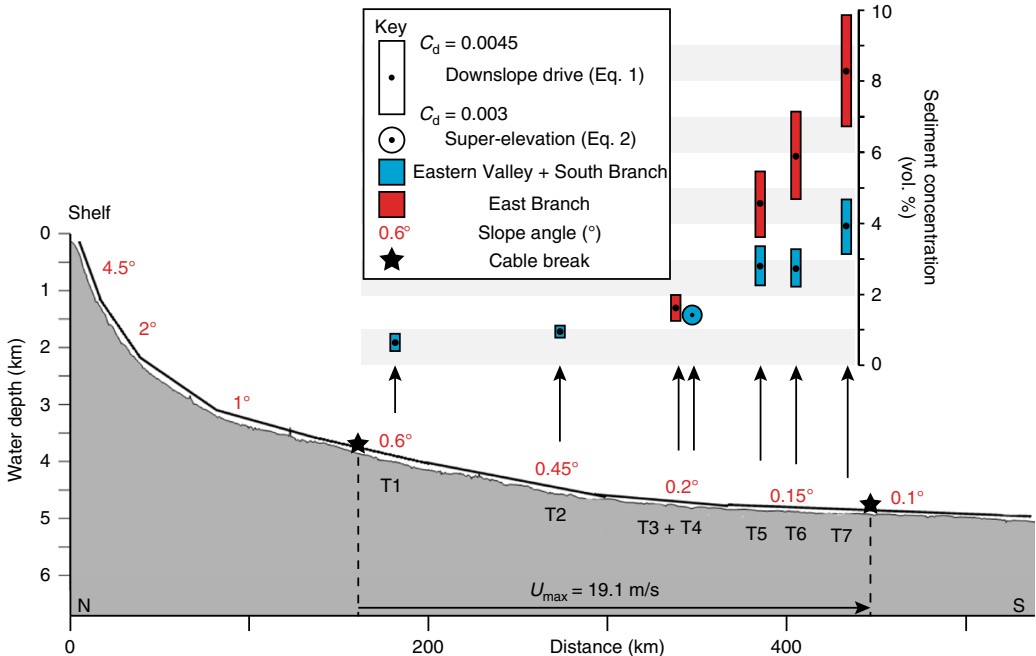

**Fig. 4** Reconstruction of the bulk sediment concentration of the 1929 flow through the Eastern Valley channel system. Parameters from Eqs. (1) and (2) are constrained from the field data, which allows sediment concentration ($C$) to be back-calculated: Velocity ($U$) from cable breaks (Fig. 1), flow thickness ($H_f$) from channel trimlines (Fig. 2) and, slope gradient ($\sin\theta$) from bathymetry (Fig. 1). Slope profile trace is shown in Fig. 1. Frictional terms ($C_d$ and $E_w$) in Eqs. (1) and (2) are not constrained by the field data and must be estimated (see 'Discussion'). This introduces error into the estimates of sediment concentration

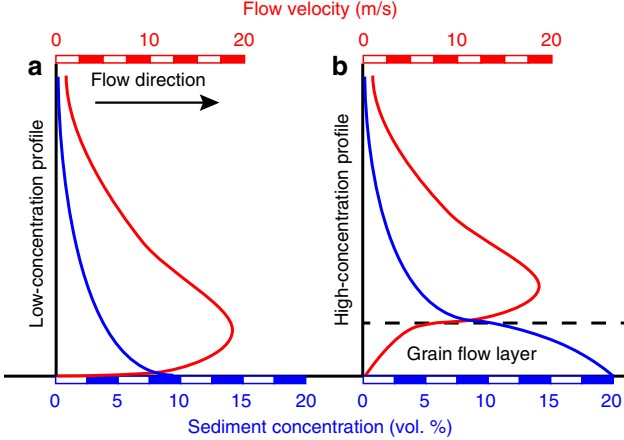

**Fig. 5** Conceptual velocity and concentration profiles vertically within a submarine gravity flow. **a** Low depth-averaged sediment concentrations result in a progressive increase in concentration towards the flow base. The shape of this profile is governed by the downward settling of grains and the upward mixing of turbulence. **b** At higher depth-averaged concentrations the flow stratifies into a hyperconcentrated basal grain flow layer overlain by a more dilute upper layer. Turbulence is suppressed in the basal layer and particles maintained in suspension via grain collisions and hindered settling effects. Our estimates of depth-averaged concentration indicate the 1929 flow probably had a high-concentration profile

Significant debate has surrounded the fundamental character of submarine gravity flows: Are they dilute and turbulent or highly-concentrated with suppressed turbulence[5, 49, 51–54]? This elementary characterization underpins how models approach predicting flow behaviour and resulting sedimentation patterns. For example, the near ubiquitously recognised 'high-concentration gravity flow' model is defined as having a variety of sediment concentrations between 5 and 9%[52], >10%[5], or 7 and 45%[55]. Without valid constraints on sediment concentration in natural flows, such models have remained conceptual.

Our estimates of depth-averaged sediment concentration for the 1929 flow of 2.7–5.4% vol. are 2–3 orders of magnitude higher compared to most small-volume dilute flows that have been measured in slope canyon systems and fjord deltas (~0.002–0.5% vol.)[6, 8, 9, 12], and exceed the upper range of sediment concentrations derived from submarine gravity flow deposits (~0.0007–2.5% vol.)[18, 19, 21, 22, 24]. Limitations in physical scaling and computational power have traditionally restricted experimental and numerical simulations to a similar range of dilute flow conditions[53, 56–58]. The few experiments that have modelled high-concentration flows have used depth-averaged concentrations in the range of 15–40% vol.[14, 36, 47, 59]. Our work shows that depth-averaged concentrations in the 1929 flow fall between these dilute and high-concentration experimental conditions. However, flow stratification likely resulted in a hyperconcentrated base (>10% vol.) and more dilute upper parts (<1% vol.). Hence, dilute and high-concentration experiments have similar concentrations to either the upper or lower layer of the flow respectively. This suggests that current scaling approaches do not appropriately relate sediment concentration to other flow properties throughout the depth of the flow. Vertical stratification may be a crucial overlooked factor in realistic modelling of natural-scale submarine flows[16, 60].

Our work provides the first validated estimates of sediment concentration for a giant submarine flow. The 1929 event had depth-averaged sediment concentrations between 2.7 and 5.4% vol. These concentrations were high enough to produce a stratified flow with a hyperconcentrated base (>10% vol.) overlain by a layer with progressively decreasing sediment concentrations. It provides a test case for scaled experiments and numerical simulations, and is a major step towards quantitative links between submarine gravity flow processes and their deposits.

## Methods

**Bathymetry and backscatter**. Bathymetry was collected aboard RV Maria S. Merian on Cruise MSM47 between 30/09/2015 and 30/10/2015. A hull-mounted Kongsberg EM122 operating at a nominal frequency of 12 kHz at a maximal swath width of 130° was used to collect bathymetric and backscatter data. Processing of the data included the application of sound velocity profiles, the application of manual and automatic methods to remove outliers and the correction for the angular dependence of backscatter intensities. Data were then gridded to a resolution of 60 m using a Gaussian weighted mean filter. Processing was done using the open source software mbsystem[61].

**Slope stratigraphy**. Sediment cores recovered stratigraphy similar to that previously documented across the Grand Banks continental slope[32]. From top to bottom: the 1929 event; a Holocene hemipelagic drape of olive-green foraminiferal ooze; a mud-dominated unit of mixed red and green turbidites of latest Pleistocene age, a thick red sticky mud unit, and stiff light grey unit of thin-bedded muddy turbidites. Deposits of the 1929 event are found in the topmost parts of the cores, overlying foraminiferal ooze or an erosion surface that cuts deeper in the stratigraphy. Using this stratigraphic framework we correlate between cores, which allows us to document the depositional and erosional record of the 1929 event as it passed through the channel system (Supplementary Fig. 2).

**Flow velocity conversions**. Direct measurements of natural small-volume submarine flows[8] show that maximum frontal velocity ($U_{max}$) is approximately five times higher than depth-averaged flow velocity ($U$). We apply this ratio to convert the frontal flow velocity of the 1929 event ($U_{max} = 19.1\,ms^{-1}$) measured by cable breaks to a depth-averaged flow velocity ($U = 3.8\,ms^{-1}$).

**Estimates of $C_d$**. The terms for frictional retardation of the flow have to be estimated using approximations from experimental and shallow water flows. The basal friction coefficient ($C_d$) is a ratio of bed roughness (skin friction and form drag) vs. the thickness of the flow passing over it. The 1929 flow has eroded and remoulded the Eastern Valley thalweg and margins, which makes it difficult to assess what sediments and bedforms were present pre-1929. Present day sediment waves found within the Eastern Valley are thought to have been formed by the 1929 flow, hence, would not have contributed to form drag. Therefore, we assume skin friction made up the majority of the basal friction beneath the 1929 flow. In channel confined turbidity currents skin friction values are generically estimated at 0.003[37]. However, from our data a more robust estimate is possible for the Eastern Valley using[62]:

$$C_d = \left[\frac{k}{B + \ln\left(\frac{Z_o}{h}\right)}\right]^2 \quad (3)$$

where, $k$ is the Von Karmen constant (0.4), $B$ is 1, $h$ is the height of the velocity maximum of the flow taken as 57 m, and $Z_o$ the bed roughness from grain size ($Z_o = 2.5 \times D_{50}/30$). Downslope through T2–T7, channel thalweg cores recover gravels with an average $D_{50}$ of 0.5–2 mm (Supplementary Fig. 2). Assuming these sediments are representative of bed roughness pre-1929, the resulting values for $C_d$ are between 0.003 and 0.0045 respectively.

**Flow duration**. The majority of the 1929 flow passed through three channel systems: the Western Valley, Eastern Valley and Grand Banks Valley[31]. The total estimated volume of the deposited sediment is ~175 km³, calculated from deposits across the lower Grand Banks slope and Sohm Abyssal Plain[30]. Assuming a packing density of 0.6, this equates to 105 km³ of sediment. If 1/3 of the flow passed through each channel system, the Eastern Valley below its confluence with Grand Banks Valley was a conduit for ~70 km³ of sediment ($V$). The cross-sectional area of the flow passing through Eastern Valley at T2 is: 23,000 m wide (channel margins) and 201 m high (trimline elevation), which is 4,634,500 m² ($A$). Flow duration = $V/UAC$. Hence, concentration estimates of 2.7–5.4% vol. result in flow durations of approximately 4 to 8 h.

**Data availability**. The datasets presented in the current study are available from the corresponding author at reasonable request.

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

## Acknowledgements

We thank the officers and crew of RV Maria S. Merian and the Shipboard Scientific Party on cruise MSM47. The cruise was supported by the Deutsche Forschungsgemeinschaft (DFG) and funding for C.J.S. was provided by the University of Leeds Institute of Applied Geoscience (IAG).

## Author contributions

C.J.S. wrote the manuscript, incorporating comments from all authors. S.K. and D.M. led the project. S.K. was chief scientist on cruise MSM47. P.F., A.G., M.S. and K.L. participated in data collection during the cruise. C.J.S. visually described cores. P.F. processed bathymetric data. D.J.W.P., A.G and M.S. contributed to data interpretation. S.K., D.J.W. P. and D.M. obtained funding for cruise MSM47.

## Additional information

**Competing interests:** The authors declare no competing interests.

