## [Peer Review File · Nature Communications]

Reviewers' comments:

Reviewer #1 (Remarks to the Author):

Review of Stevenson et al., Reconstructing the sediment concentration of a giant submarine gravity flow

Thank you for the opportunity to review the manuscript "Reconstructing the sediment concentration of a giant submarine gravity flow" by Stevenson et al. This manuscript describes the derivation of a method for calculating sediment concentrations of submarine sediment gravity flows that may help refine our understanding of the dynamics of these sediment flows and their deposit characteristics in modern and ancient settings. The authors couple existing information on the well-studied 1929 Grand Banks event with new geophysical imaging, core sampling and to evaluate transport, erosion and deposition processes primarily within the Eastern Valley channel system. Evaluation of an erosional trimline provides a key marker for evaluating thickness of the lower (higher sediment concentration) layer of the flow which is a critical observation for establishing dimensions of the flow throughout the channel system and derivation of governing equations for flow concentration. Calculated sediment concentrations are discussed in terms of the nature of the Eastern Valley sediment flows (flow structure).

The topic of this manuscript is one of great importance for the improved quantification of sediment gravity flow dynamics. Our current understanding of the dynamics, and their intra- and inter-flow variability are well known as direct observations and measurements are limited. The manuscript is well written and supported by an extensive set of what appear to be well analyzed geophysical and geological data (details of core analysis methods for the full dataset are not given). Identification and confirmation of trimline locations are well constrained within the framework of the study. There are several points that I believe could be addressed which would improve the manuscript. My recommendation falls somewhere between minor to moderate revision before acceptance.

- Although the manuscript begins with the premise that sediment concentrations of these flows have never been measured or "convincingly estimated" (line 22), it does not provide the reader with a way to assess the latter part of this statement, especially in regard to the values calculated later. How does the derived 2.5% concentration value compare to other estimates of larger canyon flows or open slope events? Why have these other estimates not been convincing? I think it is important to more broadly discuss how differing values of sediment concentration affect scaled experiments and numerical modeling. Without such a discussion, readers may be left without a way to gauge the significance of your results and their impact on the associated science.

- As the flow speeds of the Grand Banks event are well constrained by the cable breaks, it is an excellent candidate for this type of analysis...but, the complex failure mechanisms (i.e., the nature of the sediment sourcing) are also thought to be significantly different from other large submarine landslides that have more discrete source areas. Is this therefore a special case? By that I mean is this type of analysis applicable to other sediment gravity flow styles/areas? Addition of type of evaluation would be valuable in the discussion.

- How sensitive are the equations (and their results) to variations in the input values, especially those that are assumed or derived from other flow systems (e.g., specific density of sediment, basal friction coefficient, etc.)?

- I found figure 2 challenging to interpret and searched for some time to identify the dive transect and the need for the inset is not described (is it showing more cores covered by those nearby on the larger plot?). Perhaps some additional labeling would help here.

- On figure 3, there are several cases where there is not a "matching" trimline on the opposite bank (T1, T3, etc.). In these cases how is flow thickness constrained?

Jason Chaytor

Reviewer #2 (Remarks to the Author):

Dear Authors and Editor,

I have reviewed the manuscript "Reconstructing the sediment concentration 1 of a giant submarine gravity flow". The manuscript is a valid effort to use one of the iconoclastic events in sedimentary geology while improving our understanding of sedimentary processes; the topic of submarine gravity flows is intriguing and very relevant for the scientific community en large. I think the contribution needs Revisions before it can be considered for publication on Nature Communication.

I do have general philosophical comments and some specific ones I annotated going through the manuscript. I hope the Authors will find them helpful.

General comments:

The manuscript is interesting and it could be an interesting contribution. I do appreciate that the Authors used insights from recent measurements reported by Xu et al. (although Authors forgot to check out some of the more recent publications from Xu et al., with measurements from other canyons along California) and Symons et al., 2017. I do suspect that the insights gathered from these measurements might not be very helpful for the Grand Bank event. The current of the Grand Bank event where a fully-fledged ignite current traversed much of the slope; the flows recently measured in the Monterey Canyon do 'die' in the upper canyons as flows did not 'ignite'. This should not stop the Authors; my only request would be to account for that as their flow concentration could be highly underestimated while they might be overestimating flow stratification. Furthermore, hydraulic jumps, highly regarded in the original contribution from Piper et al., are somehow disregarded. Hydraulic jumps seem to almost "reset" the system at times, causing vigorous mixing and stratification disruption.

Line 26: 2.5% is given to the reader as a fact. It is estimated via the calculations (and assumptions) presented in the manuscript. Please rephrase.

Line 37: the erosion and consequent sediment entrainment is fundamental for understanding these flows. Please read Traer et al., 2012 and 2015.

Line 49: The 'trimline' approach is a valid one although not that new (used before Symons et al.). It conceptually showed its value in outcrops. The erosional thalweg is time equivalent to a depositional margin. I would invite the Authors to read the contribution from Hubbard et al., 2014.

Line 54: Data are plural.

Line 108 – onward: would the Author please check their work and compare with some of the findings of Traer et al., 2012 and 2015? I suspect you might find some interesting insights there (and possibly refine their equations).

Line 140-143: Why averaging bulk concentration here? I think spelling out a max concentration of 3.2% (and I suspect higher in part of the system) would really give a sense of these very large events.

I do think this manuscript has the potential to be a valid contribution. I hope the Authors and the Editor found the comments of some use. I listed few references for the Authors peruse.

All the best

Andrea Fildani

Hubbard, S.M., Covault, J.A., Fildani, A., Romans, B.W., 2014, Sediment transfer and deposition in slope channels: Deciphering the record of enigmatic deep-sea processes from outcrop. *Bulletin of the Geological Society of America*, 126, 5-6, 857-871.

Traer, M.M., Hilley, G.E., Fildani, A., and McHargue, T., 2012, The sensitivity of turbidity currents to mass and momentum exchanges between these underflows and their surroundings. *Journal of Geophysical Research: Earth Surface* (2003–2012) 117 (F1)

Traer, M.M., Fildani, A., McHargue, T., Hilley, G.E., 2015, Simulating depth-averaged, one-dimensional turbidity current dynamics using natural topographies. *J. Geophys. Res.: Earth Surface*, 120 (8), p. 1485-1500.

Below are specific responses to reviewer comments with their comments in black and our responses in blue:

Reviewer #1 (Remarks to the Author):

Review of Stevenson et al., Reconstructing the sediment concentration of a giant submarine gravity flow

- Although the manuscript begins with the premise that sediment concentrations of these flows have never been measured or “convincingly estimated” (line 22), it does not provide the reader with a way to assess the latter part of this statement, especially in regard to the values calculated later. How does the derived 2.5% concentration value compare to other estimates of larger canyon flows or open slope events? Why have these other estimates not been convincing? I think it is important to more broadly discuss how differing values of sediment concentration affect scaled experiments and numerical modeling. Without such a discussion, readers may be left without a way to gauge the significance of your results and their impact on the associated science.

Agreed. First, we have revised and expanded the introduction to make clearer the context of our work from the outset (Lines 32-94). This includes a revised section on previous estimates of sediment concentration, which outlines the range of previous estimates and explains why they lack validity (Lines 60-67). Second, we have revised the discussion to include a section ‘Comparison to Other Flows’ (Lines 233-260), where we compare our results to previous work and highlight the key implications for experiments. Focussing on the implications our work has for experiments has enabled us to emphasize the importance of flow stratification, which is currently not accounted for in scaled experimental approaches.

- As the flow speeds of the Grand Banks event are well constrained by the cable breaks, it is an excellent candidate for this type of analysis...but, the complex failure mechanisms (i.e., the nature of the sediment sourcing) are also thought to be significantly different from other large submarine landslides that have more discrete source areas. Is this therefore a special case? By that I mean is this type of analysis applicable to other sediment gravity flow styles/areas? Addition of type of evaluation would be valuable in the discussion.

The reviewer seems to be mixing two concepts: the landslide and the turbidity current (gravity flow). We are not sure what is the reviewer’s concern about “sediment sourcing”; the turbidity current transported a mix of landslide-derived sediment and sand eroded from the channel floor. That would be typical of most large turbidity currents initiated by landsliding (e.g the Var 1979 failure). Sediment gravity flows evolve rapidly downflow and there is little evidence that the record of initiation process is maintained as flows evolve downslope (e.g. as argued by Piper and Normark 2009). As our analysis focusses on the gravity flow part of the 1929 event, we do not consider it to be a special case.

- How sensitive are the equations (and their results) to variations in the input values, especially those that are assumed or derived from other flow systems (e.g., specific density of sediment, basal friction coefficient, etc.)?

We have added a new section in the discussion ‘Uncertainty in the Equations’ that explores key uncertainties in our equations, the most important being the basal friction and water entrainment functions (Lines 185-204).

To account for uncertainty in the basal friction coefficient we have calculated a range of values between 0.003-0.0045. This range is based on the average grain sizes found along

the channel floor and is a more appropriate estimate than a generic value from the literature (e.g. 0.003 from Komar, 1969). Revisions to the Methods have been made in line with the changes in our calculations (Lines 326-346).

The upper surface entrainment function is now discussed in light of Review 2's suggested reading and recent numerical modelling papers (Lines 198-204). The take home message from this discussion is that upper surface entrainment is likely to be 3-4 orders of magnitude less or so low as to be considered negligible. However, it is still not clear from the literature how entrainment dynamics might work in natural scale flows across a range of flow conditions. With this uncertainty, we decrease our entrainment function by 3 orders of magnitude (Line 154) and state how this effects the resulting sediment concentration estimates.

These revisions to equation parameters have resulted in slight changes to our calculated range of sediment concentrations with the addition of upper and lower limits of uncertainty. Values of sediment concentration have been revised accordingly in Figure 3 and throughout the text.

- I found figure 2 challenging to interpret and searched for some time to identify the dive transect and the need for the inset is not described (is it showing more cores covered by those nearby on the larger plot?). Perhaps some additional labeling would help here.

Extra labels have been added, which point out the dive transect and highlight the super-elevation of the trimline seen on the insert. The insert is needed because the cores in the area are closely spaced, which means core symbols overlap each other. This makes identifying each core facies difficult and obscures the underlying backscatter data. For clarity, an extra sentence has been added to the figure caption stating 'The backscatter insert shows a more detailed picture of the super-elevated trimlines around the sharp bend in South Branch (T4).' (Lines 549-551).

- On figure 3, there are several cases where there is not a "matching" trimline on the opposite bank (T1, T3, etc.). In these cases how is flow thickness constrained?

Five of the seven transects do have matching trimlines. Where field data are lacking, we have interpolated from adjacent transects using the backscatter image (Fig. 2) for the general form of the trimline. Calculations based on equation (2) use the well constrained transect T4 at the channel bend.

Reviewer #2 (Remarks to the Author):

General comments:

The manuscript is interesting and it could be an interesting contribution. I do appreciate that the Authors used insights from recent measurements reported by Xu et al. (although Authors forgot to check out some of the more recent publications from Xu et al., with measurements from other canyons along California) and Symons et al., 2017. I do suspect that the insights gathered from these measurements might not be very helpful for the Grand Bank event. The current of the Grand Bank event where a fully-fledged ignite current traversed much of the slope; the flows recently measured in the Monterey Canyon do 'die' in the upper canyons as flows did not 'ignite'.

Flows that have been measured directly, such as those in the Monterey and other California canyons, are relatively small-volume events that die within the canyons. The 1929 Event was

giant, and able to erode and transport vast amounts of sediment through the Grand Banks channel systems before depositing its sediment load over 10's thousands of square kilometres of seafloor. As the reviewer points out, it is not clear that these small-scale events would be similar to a giant event that has experienced "ignition". This is why we have refrained from extrapolating too much from these directly measured smaller-volume flows.

Specifically, we use the work of Symons et al. 2017 to support the assumption that trimlines represent the thicknesses of different parts of the flow. This is currently the only paper that provides precise trimline elevations from cores that are linked with direct measurements of flow structure.

This should not stop the Authors; my only request would be to account for that as their flow concentration could be highly underestimated while they might be overestimating flow stratification. Furthermore, hydraulic jumps, highly regarded in the original contribution from Piper et al., are somehow disregarded. Hydraulic jumps seem to almost "reset" the system at times, causing vigorous mixing and stratification disruption.

We have included a new section in the discussion that explores the uncertainty/sensitivity in our results (see also response to comments from Reviewer 1) (Lines 185-204). This has resulted in our estimates of sediment concentration increasing slightly from ~2.5 % vol. to 2.7-5.4 % vol. We have expanded the section in the discussion that explores what these depth averaged sediment concentrations might represent in terms of a vertically stratified concentration profile within the flow (Lines 217-230).

Hydraulic jumps played an important role in the initiation of the 1929 flow on the steep ($>1^\circ$) upper part of the system and influenced the vertical concentration profile of the 1929 flow as it evolved downslope. However, we do not see any sharp steps in trimline heights along the channel margins to suggest that hydraulic jumps were dramatically thickening the flow at specific locations along its pathway. Alternatively, at such high flow speeds (19.1 m/s) it is also possible that the 1929 flow was super-critical throughout the Eastern Valley Channel System. This would mean hydraulic jumps would have to have occurred at the mouths of the channels or farther out into the Sohm Abyssal Plain, which is downslope of our transects. Therefore, hydraulic jumps may not have played a role in flow stratification. No changes made.

Line 26: 2.5% is given to the reader as a fact. It is estimated via the calculations (and assumptions) presented in the manuscript. Please rephrase.

Rephrased to: 'The calculated average bulk sediment concentration of the flow was 2.7-5.4 % by volume.'

Line 37: the erosion and consequent sediment entrainment is fundamental for understanding these flows. Please read Traer et al., 2012 and 2015.

We have added a new section that discusses upper surface entrainment (Lines 198-204; see response to Reviewer 1) and another discussing how our results may reflect a spatially evolving eroding flow (Lines 206-215). In this latter section, we clarify that Equations 1 and 2 assume conservation of mass (no erosion or deposition) and that erosion is clearly observed along the flow pathway. The effects of this erosion would be to bulk the flow and increase its sediment concentration, which is a trend that is seen in Figure 4. However, the averaged flow

speed of 19.1 m/s over a changing slope will also produce this trend via Equation 1. Therefore, it is not possible to distinguish the relative contributions of these two factors in the increasing sediment concentrations seen in Figure 4. We have added several references in support of this discussion (Pantin, 1979; Parker, 1982; Parker et al., 1986; Pantin and Franklin, 2011; Traer et al., 2012, 2015).

Line 49: The ‘trimline’ approach is a valid one although not that new (used before Symons et al.). It conceptually showed its value in outcrops. The erosional thalweg is time equivalent to a depositional margin. I would invite the Authors to read the contribution from Hubbard et al., 2014.

We have added references to this statement, which have used trimlines as a proxy for flow thicknesses including: Bowen et al., 1984; Pirmez and Imran, 2003; Hubbard et al., 2014; Stevenson et al., 2014; Spsychala et al., 2017; Jobe et al., 2017.

Line 54: Data are plural.

Corrected.

Line 108 – onward: would the Author please check their work and compare with some of the findings of Traer et al., 2012 and 2015? I suspect you might find some interesting insights there (and possibly refine their equations).

We thank the reviewer for pointing us in the direction of these papers. The key insight from their work is that clear water entrainment across the upper flow interface is likely to be 3-4 orders of magnitude lower than predicted from laboratory conditions. Hence, our entrainment value (E_w) is likely to be too high. These insights also align with recent numerical modelling studies that argue submarine gravity flows have negligible upper surface entrainment and become stably stratified (Kneller et al., 2017; Luchi et al., 2017). As such we have revised our upper surface entrainment parameter (E_w) in Equation 1 to be 3 orders of magnitude smaller. This makes the value of E_w in our calculations negligible and reduces the depth averaged sediment concentration by approximately 0.03 % (up to a maximum of 1 % vol). This discussion has been added in the text (Lines 198-204) (see also responses to Reviewer 1).

Line 140-143: Why averaging bulk concentration here? I think spelling out a max concentration of 3.2% (and I suspect higher in part of the system) would really give a sense of these very large events.

The upper and lower limits of our calculated sediment concentrations are now stated (2.7-5.4 % vol.), without averaging them into a single value. Changes have been made accordingly throughout the text.

REVIEWERS' COMMENTS:

Reviewer #2 (Remarks to the Author):

Dear Authors,

Thank you for the hard work in addressing my concerns about an earlier version of this manuscript. I did find the overall contribution much improved. I do have a couple of comments/suggestions for the final draft of this manuscript.

1. In the rebuttal the Authors say: "Alternatively, at such high flow speeds (19.1 m/s) it is also possible that the 1929 flow was super-critical throughout the Eastern Valley Channel

System." I found this statement really interesting and maybe worth expanding a bit. If a flow is thoroughly supercritical for long reaches of the canyon, can it be as stratified as suggested here?

2. We were able to finally publish all the work that Dr. Traer has done for his PhD work. It did take a bit longer than hoped. Said that, I have noticed the Authors do consider a recent publication on flow stratification (Luchi et al., 2018) and I think Traer et al. 2018 might be more helpful in their case (see below the full citation). In our numerical efforts, we did find that flows do "like" finding an equilibrium, at least in the natural systems (channels) we use to constrain them.

I hope these comments do not delay further progress of this manuscript but do help clarifying the conclusions and overall message.

My regards to the Authors.

Andrea Fildani

Traer et al., 2018, Turbidity Current Dynamics: 2. Simulating Flow Evolution Toward Equilibrium in Idealized Channels. JGR-ES, <https://doi.org/10.1002/2017JF004202>

Response to reviewer comments: second round

We appreciate the constructive comments raised by reviewer Fildani, who agrees that the revisions made from the first round have improved the manuscript substantially. This second round of review follows up on some aspects of our original reviewer response. We have responded directly below. The reviewer comments are in black and our responses are in blue.

Reviewer #2 (Remarks to the Author):

Dear Authors,

Thank you for the hard work in addressing my concerns about an earlier version of this manuscript. I did find the overall contribution much improved. I do have a couple of comments/suggestions for the final draft of this manuscript.

1. In the rebuttal the Authors say: "Alternatively, at such high flow speeds (19.1 m/s) it is also possible that the 1929 flow was super-critical throughout the Eastern Valley Channel System." I found this statement really interesting and maybe worth expanding a bit. If a flow is thoroughly supercritical for long reaches of the canyon, can it be as stratified as suggested here?

Experimental evidence predicts that supercritical flows are likely to be more strongly stratified than subcritical flows (e.g. Sequeiros et al., 2010; Cortés et al., 2014). Supercritical flows have a velocity maximum lower in the flow, thus higher momentum towards the base. Velocity and concentration are linked (e.g. Felix et al., 2005) thus we'd expect more sediment towards the base of supercritical flows, and thus enhanced stratification relative to subcritical flows. Therefore, in the case of the (probably supercritical) 1929 Grand Banks submarine flow, we would expect stratification to be enhanced, not diminished. This supports our stratified flow model. No changes made.

2. We were able to finally publish all the work that Dr. Traer has done for his PhD work. It did take a bit longer than hoped. Said that, I have noticed the Authors do consider a recent publication on flow stratification (Luchi et al., 2018) and I think Traer et al. 2018 might be more helpful in their case (see below the full citation). In our numerical efforts, we did find that flows do "like" finding an equilibrium, at least in the natural systems (channels) we use to constrain them.

The contribution of Traer et al. (2018) shows that flows tend toward equilibrium flow conditions that are determined by channel morphology. Whilst an interesting paper, the key message is not entirely relevant in our case. As the reviewer pointed out in the first round of revisions, the 1929 flow probably entrained sediment along its flow path and was evolving into a more concentrated flow (i.e. not in an equilibrium state). We added a new section in response to the first round of reviews discussing this aspect of the flow (lines 191-199), supported by references of Traer et al. (2015 and 2017) as suggested by the reviewer.

The discussion surrounding upper surface entrainment is supported by references of Traer et al. (2015) as suggested by the reviewer, and by Luchi et al. (2018) and Kneller et al. (2016) who argue

from a different perspectives that upper surface entrainment is likely to be significantly less than previously estimated. Having supporting references from different research groups using different approaches strengthens the argument for reduced upper surface entrainment. Therefore, no changes made.

Stevenson and Co-Authors